



# Wind induced variability in the Northern Current (North-Western Mediterranean Sea) as depicted by a multi-platform observing system

Maristella Berta[1], Lucio Bellomo[2,3], Annalisa Griffa[1,4], Marcello Magaldi[1,5], Anne Molcard[2,3], Carlo Mantovani[1], Gian Pietro Gasparini[1], Julien Marmain[6], Anna Vetrano[1], Laurent Béguery[7], Mireno Borghini[1], Yves Barbin[8], Joel Gaggelli[8], and Céline Quentin[8]

[1]CNR-ISMAR, Lerici, Italy
[2]MIO, Université de Toulon, CNRS, IRD, La Garde, France
[3]MIO, Aix-Marseille Université, CNRS, IRD, Marseille, France
[4]RSMAS, University of Miami, Miami, FL, USA
[5]Johns Hopkins University, Baltimore, MD, USA
[6]Degreane Horizon, Cuers, France
[7]ACSA, 9 Europarc, 13590 Meyreuil, France
[8]MIO, CNRS, Aix-Marseille Université, Université de Toulon, IRD, Marseille, France

Correspondence: Maristella Berta (maristella.berta@sp.ismar.cnr.it)

**Abstract.** The variability and evolution of the Northern Current (NC) in the area off Toulon is studied for two weeks in December 2011 using data from a glider, a HF radar network, vessel surveys, a meteo station, and an atmospheric model. The NC variability is dominated by a synoptic response to wind events, even though a seasonal trend is also observed, transitioning from late summer to fall-winter conditions. With weak winds the current is mostly zonal and in geostrophic balance even at the surface, with a zonal transport associated to the NC of $\approx 1$ Sv. Strong westerly wind events (longer than 2-3 days) induce an interplay between the direct wind induced ageostrophic response and the geostrophic component: upwelling is observed, with offshore surface transport, surface cooling, flattening of the isopycnals and reduced zonal geostrophic transport (0.5-0.7 Sv). The sea surface response to wind events, as observed by the HF radar, shows total currents rotated at $\approx -55°$ to $-90°$ to the right of the wind. Performing a decomposition between geostrophic and ageostrophic components of the surface currents, the wind driven ageostrophic component is found to rotate of $\approx -25°$ to $-30°$ to the right of the wind. The ageostrophic component magnitude corresponds to $\approx 2\%$ of the wind speed.



# 1 Introduction

The Liguro-Provenço-Catalan Current, also called Northern Current (Millot, 1999), is a boundary current corresponding to the upper limb of the Western Mediterranean circulation (Fig.1a). It originates in the Ligurian Sea due to the convergence of the two currents flowing along each side of the Corsica island (Astraldi and Gasparini, 1992), namely the Western and Eastern
Corsica Currents (WCC and ECC). The Northern Current (denoted NC hereafter) flows southwestward along the continental slope of the Ligurian-Provençal and Balearic basins with a certain degree of continuity and may thus be recognized as a single entity as far as the Catalan Sea (Millot, 1987; Font et al., 1988; Garcia et al., 1994). Its importance mainly relies on the fact that all water masses in the area - namely the Modified Atlantic Water (MAW), the Levantine Intermediate Water (LIW) and the Western Mediterranean Deep Water (WMDW) - are carried by the current during its flowing (Conan and Millot, 1995). As
a result, the NC is known to influence the coastal circulation of the Gulf of Lion (Duchez et al., 2012) and most importantly, to modulate the supply of salt and/or heat by lateral advection in the convection areas (the so called preconditioning, see Schroeder et al. (2010)) and thus affect the important deep water formation process in the Western Mediterranean. Also, the NC hugs the highly populated coasts of Italy and France, where areas of intense industrial development alternate with turistical and environmental relevant Marine Protected Areas (MPAs). Understanding the flow dynamics and how it carries biological
and pollutant quantities is of great importance for a correct management of coastal and marine activities.

The NC has been intensively observed in a few specific places of the southern French coast, namely a) off Nice (Béthoux et al., 1982; Taupier-Letage and Millot, 1986; Béthoux et al., 1988; Albérola et al., 1995a; Sammari et al., 1995); b) at the eastern edge of the Gulf of Lion, off Marseilles (Conan and Millot, 1995; Flexas et al., 2002; Albérola and Millot, 2003; Forget et al., 2008) and c) along the Gulf of Lion shelf break (Lapouyade and de Madron, 2001; Petrenko, 2003; André et al.,
2009; Rubio et al., 2009). All this past literature agrees on some of its large scale characteristics: the NC is confined within approximately $35 - 40$ km from the coast and has an average geostrophic transport value of about 1.8 Sv (1 Sv = $10^6$ m$^3$ s$^{-1}$) (Béthoux et al., 1982; Sammari et al., 1995). It exhibits a marked annual cycle: during winter it gets stronger, closer to coast, narrowing (about 25 km) and deepening (about 450 m), while in summer, it weakens, extends more offshore increasing its width around 40 km, and gets thinner (about 250 m). Mesoscale activity also increases from late autumn to winter and then it
decreases in summer, and it is concentrated in two main time windows of 3-6 days and 10-20 days respectively (Taupier-Letage and Millot, 1986; Millot, 1987; Albérola et al., 1995a; Sammari et al., 1995).

Despite the numerous articles above, the study of the NC and its variability still represents an active area of research, especially at scales shorter than seasonal. Transport measurements provide a wide range of values in the literature, reaching minimum values of 0.5 Sverdrup and showing significant variability in time (Conan and Millot, 1995) and in space (Petrenko,
2003). The reason for this variability within the annual cycle is not clear yet. It has been suggested that it may be due to different freshwater signals from precipitation and river runoff (Béthoux et al., 1988), to the WMDW formation process (Crépon and Boukthir, 1987), or inherited by the distinct behavior of the WCC and the ECC, whose annual peaks differ of several months (Astraldi and Gasparini, 1992).



In particular, the study of variability of the NC in the area off Toulon (between Nice and Marseilles) deserves particular attention for various reasons. The NC dynamics has been undersampled in the past and far less documented than in other regions such as off Nice and Marseille. Only recently the NC mesoscale meandering off Toulon has been specifically investigated through the comparison of models and observations (Guihou et al., 2013) and with data assimilation (Marmain et al., 2014).

These studies indicate the role of prevailing northwesterly winds in the modulation of the NC circulation. In fact, the area near Toulon is peculiar because the mean NC circulation is resisted by the prevailing northwesterly winds in the area. The upwelling prone winds can significantly alter the current, to the point that in some occasions they can stop its westward propagation and control its penetration in the Gulf of Lion (Millot and Wald, 1980). When this happens, a frontal zone separating warm NC waters and cold waters upwelled from the Gulf of Lion is established near the coast, and it has clearly been observed from

satellite images (Millot and Wald, 1980).

In this paper, the variability of the NC in the area off Toulon is studied using results from a multi-platform experiment involving HF radar, glider, a meteo station, and a research vessel, carried out in the framework of the EU-MED project TOSCA (Tracking Oil Spills and Coastal Awareness network, http://www.tosca-med.eu). The experiment took place during a period of approximately two weeks in December 2011 (December 2-19), characterized by intense westerly and northwesterly wind

events. The wind response of coastal boundary currents such as the NC is complex, and based on the interplay between the direct ageostrophic response in the surface layers and the geostrophic modifications occurring along the whole water column. The response of boundary currents to winds has been studied by several authors using numerical model results, satellite and in-situ data (Magaldi et al., 2010; Aguiar et al., 2014; Schaeffer and Roughan, 2015). Experimental works are mostly based on a statistical approach, i.e. using long time series of wind and currents and computing correlations or identifying the main flow

patterns corresponding to specific wind forcing (Kosro, 2005; Kim et al., 2010; Mihanović et al., 2011; Yuan et al., 2017).

Here, we focus on the dynamics of specific wind events, and take advantage of the multi-platform information, especially from HF radars for the surface and glider for the water column. Our general goal is to identify the main processes at work, and to investigate how to unravel the interplay between the direct wind induced surface response and the geostrophic response. The following two specific goals are pursued:

1) to describe the sea current system evolution during the period of interest, quantifying the water column response to the wind in terms of isopycnal and zonal geostrophic velocity evolution;

2) to investigate the surface layer response to the wind, attempting a decomposition between geostrophic and ageostrophic components of the velocity.

The problem of decomposing the surface flow in geostrophic and ageostrophic components has been previously addressed

in several works. Earlier works used hydrographic or ADCP information to infer the geostrophic flow (e.g. Weller et al. (1991); Chereskin and Roemmich (1991); Wijffels et al. (1994)), while more recently the global altimetric products (Lagerloef et al., 1999) have been used in several applications. Rio and Hernandez (2003) used SVP drifters and altimetry to infer information at global scales, while HF radar results with altimetry have been used to compute wind driven velocities in the Kuroshio Current (Tokeshi et al., 2007) and in the South Atlantic Bight (Yuan et al., 2017). For coastal boundary currents such as the NC and for

the space and time scales we are interested in, though, the use of altimeter is not appropriate. In this paper, we investigate the



use of glider data in conjunction with HF radar. There are several challenges in using glider transects, including limited space and time sampling (Piterbarg et al., 2014) and the fact that only one component of the geostrophic velocity can be retrieved. Here we investigate how to best combine glider and HF radar data and in which flow conditions they can be best exploited.

The paper is organized as follows. In *Section 2*, the main questions addressed in the paper are stated, and the main definitions are introduced. In *Section 3*, a description of the data as well as the methods used to analyze them is provided. In *Section 4*, a description of the sea current system evolution over the whole water column is provided, while the analysis of the wind response in the surface layer is performed in *Section 5*. A summary and discussion are provided in *Section 6*.

## 2   Background, statement of the problem and definitions

As stated in the Introduction, the general goal of this work is to investigate the variability of the NC during the period of interest, with special focus on the response of the system to wind forcing, using information from glider and HF radar. The velocity field of the system $\mathbf{u}(x,y,z,t)$ can always be written through kinematic decomposition as the superposition of a geostrophic and an ageostrophic component:

$$\mathbf{u} = \mathbf{u}_g + \mathbf{u}_a \tag{1}$$

The geostrophic part $\mathbf{u}_g$ obeys to the balance between pressure gradient and earth rotation (Coriolis). In the case of a boundary current, $\mathbf{u}_g$ is expected to be driven by the large scale pressure gradient of the general circulation, as well as by the more localized pressure gradient of mesoscale phenomena such as meanders and eddies (Centurioni et al., 2008; Gangopadhyay et al., 2013). The ageostrophic component, $\mathbf{u}_a$ includes the phenomena that are not in geostrophic balance. In the water column interior, we can expect that the flow is mostly geostrophic at scales longer than a day, while high frequency processes such as internal waves, tides or inertial oscillations, play an important role at shorter time scales (Mensa et al., 2013).

At the surface, in addition to high frequency processes, also ageostrophic processes related to direct wind and air-sea interaction are relevant. In particular, the surface velocity, $\mathbf{u_S}(x,y,t)$ can be written as:

$$\mathbf{u_S} = \mathbf{u}_{Sg} + \mathbf{u}_{SaW} + \mathbf{u}_{SR} \tag{2}$$

where $\mathbf{u}_{Sg}$ is the surface geostrophic velocity, $\mathbf{u}_{SaW}$ is the directly wind driven component, and $\mathbf{u}_{SR}$ includes all the other high frequency processes such as tides, inertial and submesoscale variability, that for the purpose of this paper we will refer to as "residual".

The wind response $\mathbf{u}_{SaW}$ has been studied for many decades, starting from the pioneering work by Ekman (1905), considering idealized solutions of the balance between Coriolis and friction. The classic Ekman solution, valid in stationary and homogeneous conditions of infinite domain in the horizontal and at depth, is characterized by a surface current at $45°$ to the right of the wind (in the Northern hemisphere) spiraling with depth in the surface layer. The solution is highly dependent





on the specific parameterization of wind stress and vertical diffusivity. As an example, assuming that the wind stress linearly decreases with depth, leads to the "slab" solution, which is $90°$ to the right of the wind and constant with depth (Pollard and Millard, 1970). The choice of other parameterizations as well as the presence of boundaries, finite depth, time dependence and inhomogeneity further modifies the solution (Endoh and Nitta, 1971; Ralph and Niiler, 1999; Crise et al., 2006). Also, it

can be expected that the simplified balance of the Ekman equation only partially captures the dynamics in realistic conditions, because of the interactions between the various processes. Indeed, direct measurements show a great range of variability in the observed wind response (Rio and Hernandez, 2003; Sentchev et al., 2017). In summary, even though the idealized solutions provide very important general guidance, the actual surface response to the wind in realistic conditions is still an open question (Stanichny et al., 2016), and it can be expected to depend on the specific environmental conditions.

We notice that there is a well known direct dynamical link between $\mathbf{u}_{SaW}$ and the geostrophic velocity $\mathbf{u}_g$. In the case of a boundary current, when the wind induces cross-shore transport, the cross-shore pressure gradient can be modified provided that the wind acts for a sufficiently long time scale, $T_W$ (Whitney and Garvine, 2005). For the Northern Current, Piterbarg et al. (2014) estimated that for winds of the order of $10\ ms^{-1}$ $T_W$ is of the order of 2.5 - 3 days. In the case of upwelling prone winds, as is the case of the westerly winds considered here, the ageostrophic surface response causes offshore transport that is

compensated by deep water coastal upwelling, modifying both the surface pressure gradient and the water column stratification estimated by the potential density anomaly $\sigma(z)$. This in turns, alters the along-shore geostrophic component of the flow. In addition to this main mechanism, the geostrophic response can be further modulated by several other mechanisms, such as nonlinear frontal wind response (Oguz et al., 2017), and interactions with mesoscale and submesoscale instabilities.

The work performed here has two specific goals:

1) Describe the variability of the NC over the whole water column and investigate its relationship with wind forcing.

The full 3D description of $\mathbf{u}(x,y,z,t)$ is clearly not available from our data, but the data from glider and HF radar provide a good first approximation of the sea current system. The radar data provide estimates of the 2-D surface velocities $\mathbf{u}(x,y,t)$, while the glider data provide information on the interior flow in terms of stratification, $\sigma(y,z,t)$, and zonal geostrophic velocity, $u_g(y,z,t)$, estimated along the cross-shore glider transect. In this part of the work, we concentrate on variability over time scales

of one day or longer, in keeping with the use of glider transect data and geostrophic velocities. In particular, we investigate the response of the sea current system during intense westerly wind events and concentrate on the main mechanism between wind induced surface transport and upwelling response. Other mechanisms such as nonlinear wind response and interaction with instabilities are not directly considered here because we do not have enough information to resolve the various aspects.

2) Further investigate the wind response in the surface layer and test a method to estimate the induced ageostrophic compo-

nent $\mathbf{u}_{SaW}$.

The high frequency response of the surface velocity $\mathbf{u_S}(x,y,t)$ as provided by the radar is used as a basis for the analysis. The estimate of $\mathbf{u}_{SaW}$ is performed during periods of high winds, when it can be assumed that this term is prevalent with respect to $\mathbf{u}_{SR}$ in eq. (2). Still, the difficulty lies in identifying a reliable estimate for $\mathbf{u}_{Sg}(x.y.t)$ to be subtracted to $\mathbf{u_S}(x,y,t)$ in eq. (2). A simple method based on the combined analysis of glider and HF radar data is put forth, valid for specific dynamic

regimes.



## 3 Data and methods

Here we review the data sets used in this study and describe the main data treatments and analyses that have been carried out. The data spatial coverage is shown in Fig.1b, while a time line of the measurements is shown in Fig.2a

### 3.1 Vessel based measurements

During the experimental effort, an oceanographic cruise took place aboard the Italian R/V Urania between December 10 and 15, 2011. While measurements were performed within an extended offshore area between Nice and Toulon, here we consider only the Toulon measurements taken on December 13 and 14. A total of 11 CTD stations were sampled in the surrounding of the glider transect (Fig.1b). The CTD data have been used for intercomparison and calibration of the glider hydrographic data as further discussed in *Sections 3.2-3.3*.

### 3.2 Gliders

An autonomous underwater glider of the Slocum kind (Jones et al., 2005), manufactured by Webb Research Corporation, has been deployed and maintained operational during the period December 2-19 (Fig. 2a). The glider, named Hannon, covered 6 repeated meridional transects off Toulon extending ∼70 km offshore (Fig. 1b). The onshore half of the transects laid within the HF radar coverage. The maximum profiling depth was set to 1000 m, giving mean horizontal distance between consecutive

profiles, mean horizontal speed, and mean vertical speed of 1.7 km, 35 cms$^{-1}$, and 20 cms$^{-1}$, respectively. The timing details of each transect are summarized in the timeline in Fig.2a and in Table 1.

Hannon was equipped with an unpumped SBE 41 CTD manufactured by Sea-Bird Electronics. CTD data were processed with dedicated Matlab routines taking care of all classic CTD response times and alignment issues, and all parameters were rebinned onto a regular vertical grid with a step of 2 dbar. Due to the use of an unpumped CTD and to the variable speed of the

autonomous vehicle, the corrections were made speed-dependent using the glider speed computed through pressure variations and tilt angle. The thermal lag of the conductivity sensor was dealt also with speed-dependent coefficients (Morison et al., 1994) experimentally found through the technique proposed by Garau et al. (2011). The optimum values of the coefficients were $\alpha_o = 0.5447$, $\alpha_s = 0.0708$, $\tau_o = 9.5117$, and $\tau_s = 7.69$. Finally, temperature and conductivity values were post-calibrated against the Sea-Bird Electronics SBE 911+ CTD deployed from the R/V. For this purpose, only glider and ship profiles distant

less than 14 km and separated by less than 12 h were considered, discarding all data shallower than 600 dbar.

### 3.3 Glider data analyses: isopycnal structure and computation of zonal geostrophic velocities

The glider hydrographic data were used to describe the NC current stratification and evolution. The hydrographic glider data have also been used to compute relative geostrophic velocities in the direction perpendicular to the glider transects, corresponding to the zonal direction. Potential density profiles were used, previously low-pass filtered through a Gaussian filter

with 9-km cut-off wavelength (Rossby radius found through the dynamic mode decomposition of the average Brunt-Väisälä frequency profiles (Kundu et al., 1975). Although depth-averaged velocities from the glider could have been used to reference





the geostrophic velocities (as done e.g. by Davis et al. (2008)), a calibration problem in Hannon magnetic compass prevented us from doing so. Therefore, the velocities were referenced to a level of no motion $z_0$, assumed to be in the range between 500-700 m. Sensitivity tests on $z_0$ between 500 and 700 show very limited variability, with root mean square differences in the mean upper layer velocity of $\approx 2\%$. In the following, results with $z_0 = 500$ m are used.

The zonal geostrophic velocities were used also to compute the integrated transport, using the 5 cms$^{-1}$ isotach to identify the Northern Current, as done e.g. by Albérola et al. (1995a) and Conan and Millot (1995).

### 3.4  HF radar

The HF radar system has been operational during the period December 6-19 (Fig. 2a), covering the area in front of Toulon (Fig. 1b). The HF radar installation is based on the WERA technology (Gurgel et al., 1999) and relies on two systems. The first

one (Fort Peyras, "FP" in Fig. 1b) has a quasi-monostatic configuration with an irregular, W-shaped 8-antenna receiving array and 2 monopoles performing the emission while forming a zero in the direction of the receiver. The peculiarity of the receiving array geometry is imposed by the environment of the site, a dismissed military base. The second system (Cap Bénat, "CB" in Fig. 1b) has a fully bistatic configuration, with the 2 monopoles in FP employed as emitter, and a regular linear 8-antenna array in CB operated as receiver (transmitter and receiver are 35 km apart).

The two systems operate at a frequency of 16.1 MHz with 50 kHz bandwidth, giving a nominal range resolution in the radial direction of 3 km. Antenna patterns are routinely measured almost every year and they had been applied to the December 2011 data set. The azimuthal processing is done with the MUSIC (MUltiple SIgnal Characterization) direction finding algorithm with a nominal 2 deg spacing (Lipa et al., 2006; Molcard et al., 2009; Sentchev et al., 2013), and radial velocity maps are produced every 20 min by integrating over the previous hour. Total velocity maps are then obtained on a regular 2 km grid

through a local interpolation method which, at each grid point, minimizes the Mean Square Error (MSE) between the projection of the total velocity onto the radial directions and the radial velocities available within a 3 km-radius circle (Lipa and Barrick, 1983). Classically, total velocities are only computed when the angle between radial data from the two systems lies within the range $30 - 150°$, which corresponds to GDOP values smaller than 2.5 (Chapman et al., 1997).

In our case, the requirement on the angle had to be reduced to $20 - 160°$ (corresponding approximately to GDOP values

smaller than 4) in order to keep an acceptable offshore coverage on the region for the bistatic configuration. The Toulon radar system has been validated also during other TOSCA experiments involving the deployment of drifters, used to compare HF radar fields and derived velocities from in situ trajectories. Results show a high level of accuracy, on average 80% agreement between drifters and HF radar, consistently using FP as quasi-monostatic configuration, CB as a different bistatic configuration (with the emitter in Porquerolles transmitter and receiver are 16 km apart) and GDOP values smaller than 2.5 (Berta et al.,

2014; Bellomo et al., 2015).

HF radar measured velocities are the results of a vertical integration, through an exponential weighting function, over a characteristic depth $\lambda_0/4\pi$ (Stewart and Joy, 1974), where $\lambda_0$ is the resonant Bragg wavelength, which is half the wavelength of the emitted electromagnetic wave. For our systems, operating at a central frequency of 16.1 MHz, the characteristic depth is about 75 cm.





Due to the proximity of the FP site emitter with respect to the receiver array, imposed once again by the constraints of the military base, the HF radar coverage systematically experienced a drastic loss under strong Mistral winds. In fact, the wind-induced vibrations of the emitter antennas generated a phase noise in the receiver antennas frequency spectrum, making in turn the signal-to-noise ratio unacceptably low at usually well-covered distances. Unfortunately, the problem was solved only after
the observational campaign by placing the emitter farther outside the military base.

### 3.5 Analysis and interpretation of HF radar data

The HF radar fields are used to describe the evolution of the surface velocities. Preliminary tests have been performed using the raw data as well as low pass filtered data with a cut-off period of 36 h. Results, using raw and filtered data, agree within 86% to 99% in terms of average and rms velocites for $v$ and $u$ component respectively, so that only results based on raw data
are presented in the following.

We point out an important general issue regarding the interpretation of results based on HF radar velocities. Several papers (Mao and Heron, 2008; Ardhuin et al., 2009) have pointed out that the HF radar based surface velocity has, in addition to the actual Eulerian velocity $\mathbf{u}(x,y,t)$, also a nonlinear wave correction (Weber and Barrick, 1977) that can be interpreted as a filtered surface Stokes drift, including the contribution of waves with wavelengths longer than the Bragg resonant wavelength.

A method to estimate this term has been proposed based on an accurate numerical wave model (Ardhuin et al., 2009). A debate is ongoing in the literature regarding the magnitude of the term and whether is significant or not (Mao and Heron, 2008; Röhrs et al., 2015). An important factor is the fetch, since the longer is the fetch the more important is the contribution of long waves (Essen et al., 2000). In our case, the Bragg resonant wavelength for our system is approximately 9 m, and the Stokes-like term cannot be explicitly estimated following Ardhuin et al. (2009) because we do not have access to the full wave spectrum
information for the period of interest. Since the wind is predominantly from the west and north-west and given the geography of the region (Fig.1a), we can expect that the fetch is limited and therefore the term is unlikely to be relevant. In absence of a quantitative estimate, though, we caution that the wind response term $\mathbf{u}_{SaW}$ for our measurement could include not only the Ekman-type Eulerian response, but also a possible Lagrangian contribution from Stokes drift. We will come back to this point in *Section 5*.

### 3.6 Wind data from meteo station and model

Hourly wind speed and direction measurements have been provided for the complete period of 2-19 December by the MeteoFrance's Porquerolles meteo station whose location is shown in Fig.1b. In addition, Meteo France's ALADIN operational regional model (1/10° and 3 h space and time resolution, respectively) has also been used.

Time series of wind speed and directions from the meteo station and from the model averaged over the area of interest
(Fig.1b) are shown in Fig.2b. The results are qualitatively very similar, showing the good agreement of the model with the data and indicating that the wind patterns in the area are not characterized by strong gradients during the period of interest. Only during the last few days, after December 18, the two time series show significant differences.



From the wind time series, events of high wind speed have been identified, considering a threshold of $10 \text{ ms}^{-1}$. This choice is consistent with the Beaufort scale (www.spc.noaa.gov), and it is confirmed a-posteriori by the consistency of the results as discussed in *Section 5*. As shown in Fig.2a,b, the period of interest is characterized by two main events that last more than 3 days, E1 and E3, and a shorter event lasting less than 1 day, E2. The main wind direction is westerly and northwesterly, compatible with Mistral events in the area. We notice that, E1 and E3 have duration longer than the estimated value of $T_W$ in the area (2-3 days), and therefore are likely to influence stratification and geostrophic velocity in the NC, as discussed in *Section 2*. The opposite holds for E2.

## 3.7 Summary of the measurements

In summary, the time line of the main measurements carried out during the experiment is provided in Fig. 2a and includes:

- Glider measurements (red boxes). They cover the period December 2-19 for a total of 6 transects, alternating offshore and inshore routes.

- HF radar measurements (solid gray). They start December 6, so that the first glider transect does not have contemporary HF radar data. The periods in which the glider transects fall inside the radar coverage are shown by green dashed lines.

- Wind measurements and model outputs. They are available during the whole period December 2-19. The three wind events E1, E2, and E3 are shown as dashed black lines.

## 4 Water column stratification and geostrophic variability, and effects of wind forcing

In the following, the variability of the Northern Current is described during the period of interest December 2-19. For simplicity we partition the time of the analyses following the same time intervals as for the glider transects (Table 1, Fig.2a). For each transect period, we provide a basic description based on the wind evolution together with the glider results on hydrography and zonal geostrophic velocity, and (when available) on the time-averaged surface velocity fields from HF radar. Radar velocity field averages are computed during periods in which the glider transects fall inside the radar coverage and considering only grid points with more than $80\%$ measurement coverage in time, in order to avoid mean flow contamination due to inhomogeneous coverages. For selected transects, all the information above from glider and HF radar are shown grouped together in a single multi-panel figure (Figs. 3-6).

In order to quantify the flow variability we show also some comprehensive figures that compare all transects in terms of the following metrics: evolution of the isopycnals $\sigma(z)$ (Fig.7), zonal geostrophic transport (Fig.8), and comparison between surface zonal geostrophic velocity $u_{Sg}$ estimated from the glider data and total surface velocity $\mathbf{u_S}$ considering HF radar data along the glider tracks (Fig.9).





### 4.1 Transect 1 : initially calm condition followed by the onset of wind event E1

During Transect 1 (December 2-4) the glider moves offshore from the coast, and no HF radar are available (see Fig.2a). The wind speed (Fig.2b) is initially weak, and then the first westerly event E1 starts on December 3. Notice that the period before the experiment (not shown) was characterized by calm conditions, with a week of weak winds, with speed smaller than 7.5

$ms^{-1}$.

The hydrographic properties (potential temperature $\theta$ and practical salinity $S$) of Transect 1 are shown in panels a,b of Fig.3 for the first 300 m, with overlying isolines of potential density anomaly $\sigma(z)$. The transect plot origin of all hydrographic panels shown in the following figures is located a few kilometers off the coast (starting point of the glider mission). Following Boucher et al. (1987), we refer to the three zones that can be typically distinguished in the NC structure on the basis of the

shape of the isopycnals with potential density anomaly $\sigma$ in the range 28.7 - 29.05 $kgm^{-3}$: the inshore flat part of the isopycnals defines the "coastal" (or marginal) zone, the sloping part identifies the "frontal" zone where the NC is most energetic, and the flat offshore part denotes the "central" basin zone. Here, we use $\sigma$=28.7 $kgm^{-3}$ to characterize these zones (see also Fig. 7). The coastal marginal zone (typically flat) is not visible in the glider transect, while the frontal zone is evident and it extends up to $\approx 42.5°$ N, i.e. at $\approx$ 40-45 km offshore. The $\theta$ and $S$ transects (Fig. 3a,b) show a strong thermocline at 50 m in the

offshore central region, accompanied by the presence of a salinity minimum right below it in the frontal zone. These conditions are typical of late summer conditions (Guibout (1987) pp. 16 and 18 off Toulon and p. 36 off Nice), that apparently are still present at the beginning of the experiment. The zonal geostrophic velocity $u_g$ computed along the transect shows a westward current extending up to 60 km off the coast with core (up to 0.4$ms^{-1}$ in first 100m) located at $\sim$42.7°-42.8° (Fig.3c). The offshore limit and depth of the NC are approximately $\sim$ 55 km and $\sim$175 m, respectively, with the core situated roughly at km

25 (Albérola et al., 1995a; Petrenko, 2003). The relatively large width and shallowness of the observed current here, known to be narrower and deeper than this during winter Albérola et al. (1995a), confirm the presence of late-summer conditions at the beginning of the experiment.

### 4.2 Transect 2-3 : Wind event E1

During the following two Transects, 2 and 3, (December 4-7 and 7-9 respectively) while the glider travels back inshore and

then offshore again, the wind is mostly dominated by the westerly event E1 (Fig.2b), tapering off toward the end of the period. HF radar data are available starting from December 6 (Fig.2a). The results of the two transects are qualitatively similar, so the complete results are shown only for Transect 3 in Fig.4.

The hydrographic transects from the glider show a significant change with respect to Transect 1 (Fig.3). Surface waters in the frontal zone are colder of $\approx 1.5°$C and the shape of the isopycnals is significantly flattened, while the mixed layer in

the offshore central part is deepened of $\approx$ 20-30 m. This can be seen clearly also in Fig.7, where the isopycnal with $\sigma = 28.7$ $kgm^{-3}$ for all transects are shown. Transects 2 and 3 show very similar isopycnal shape in the frontal area, flatter and shallower than for Transect 1. In the offshore central region, on the other hand, the isopycnal of Transect 2 is similar to the one of Transect 1, while for Transect 3 it deepens of $\approx$ 20-30 m. This is likely due to the different sampling times, since the glider covers the



offshore region in Transect 3 about two days later than for Transect 2, and Transect 2 itself takes place shortly after the onset of E1 wind event (Fig.2a).

The average surface velocity $\mathbf{u_S}$ depicted by the HF radar during the glider sampling (Fig.2a) is shown in Fig.4d. Notice the reduced coverage with respect to the expected coverage (Fig.1b), due to wind induced interferences as discussed in *Section 3.4*.

The velocity shows an overall offshore transport, with localized eastward reversals of the zonal current in the north western part of the radar coverage. The prevalent offshore current is consistent with the expected wind driven transport to the right of the prevalent westerly wind (more in depth discussion is provided in *Section 5*).

Overall, the results suggest the presence of an upwelling phenomenon associated to the offshore transport induced by the westerly winds, that causes the flattening of the isopycnals in the frontal region and the cooling at the sea surface. In addition to

the upwelling, other phenomena related to wind response are likely to occur, as suggested by the offshore isopycnal deepening in Figs.4 and 7, likely due to mixing that deepens the thermocline and possibly to convection processes.

The zonal geostrophic velocity $u_{g(y,z)}$ computed from the glider (Fig.4c) is mostly westward but reduced with respect to Transect 1 (T1) in Fig.3c, as expected during an upwelling episode. The difference between geostrophic velocity in T1 and T3, considering also the elapsed time between the two transects, is consistent with the estimate of the time scale $T_W$ (Whitney and

Garvine, 2005) of 2-3 days needed for wind events to affect geostrophy in the NC (Piterbarg et al., 2014), and it suggests that E1 can modify stratification and geostrophic velocity after a sufficiently long wind forcing period.

The change in $u_{g(y,z)}$ is quantified computing the corresponding zonal transport (Fig.8). For the integrated transport computation the 5 cms$^{-1}$ isotach was used to identify the Northern Current, as done e.g. by Albérola et al. (1995a) and Conan and Millot (1995). A very strong variability is seen with respect to the first transect, with the transport reduced of $\approx 50\%$ going

from $\approx 1.15$ Sv for T1 to 0.7-0.6 Sv for T2 and T3.

Finally, we provide a preliminary assessment of the surface deviation from geostrophy (that will be further investigated in *Section 5*) by comparing the magnitude of the zonal geostrophic velocity $u_{Sg}$ from the glider with the total velocity $\mathbf{u_S}$ depicted by the radar along the glider track (Fig.9). During T2, the zonal radar total velocity is significantly lower than the geostrophic one, while the meridional component of the total velocity is almost the double of the zonal one. Results for T3 are qualitatively

similar but incomplete because of the limited radar coverage. The observed magnitude difference in the zonal component from glider (geostrophic) and radar (total velocity) suggests that ageostrophic processes play a non negligible role at the sea surface for T2-T3.

### 4.3 Transects 4-5 : calm conditions interrupted by wind event E2 and onset of E3

The following two transects, T4 and T5, (December 9-12, 12-15 respectively) are characterized by mostly calm wind conditions

(Fig.2a), except for the brief westerly wind event E2, occurring in between the two transects. Winds during the calm period are mostly westerly, except for some strong direction oscillations prevalent for lowest wind speed (below 4-5ms$^{-1}$), mostly evident in the Porquerolles station records. During the last day of T5, the onset of the third wind event E3 occurs. Notice that E2 lasts less than a day, which is significantly less than $T_W$ estimetad by Piterbarg et al. (2014), so that it is not expected to



modify the stratification and geostrophic velocity, structure even though of course it influences surface velocity. The results of T4 and T5 are qualitatively similar, and the T5 results are shown in Fig.5

The hydrographic transects (Fig.5a,b) indicate a narrowing and steepening of the frontal region characterized by warmer and fresher water in the surface layer. This is shown also by the $\sigma$ isopycnals in Fig.7 that are very similar for T4 and T5, and

suggests a strong frontal area closer to the coast, approximately north of $42.7°$ N.

The zonal geostrophic velocity $u_{Sg}$ (Fig.5c) indeed shows a narrower but deeper and intensified current. The corresponding zonal geostrophic transport (Fig.8) increases with respect to T2-T3, reaching values of $\approx 1 - 0.9$ Sv for T4 and T5.

The radar surface velocity (Fig.5d) is very different from the previous transects (Fig.4d). The coverage is more extended and the velocity is almost completely zonal, showing the typical westward structure of the Northern Current. Notice that, even

though the current actually deviate from the zonal direction during E2 (as further discussed in *Section 5*), the contribution to the average shown in Fig.5d is very modest, because the average is performed during the glider sampling that overlaps only for a few hours with E2 (Fig.2a). Also, E2 does not appear to modify significantly the stratification, the geostrophic velocity and the associated transport, as it can be seen by the comparison of the two transects T4 and T5 in Figs. 7-8. This is in keeping with the fact that the duration of E2 is shorter than $T_W$.

The surface comparison between geostrophic $u_{Sg}$ and total velocity $\mathbf{u_S}$ from HF radar shows an excellent agreement in the zonal direction for both T4 and T5 (Fig.9c,d), while the meridional component in the frontal region is significantly lower than the zonal one (approximately $70\%$ and $85\%$ for T4 and T5 respectively). Overall, the results suggest that the current during these two transects is mostly zonal and geostrophic even at the surface, i.e. sustained by the large scale pressure gradient that maintain the general and mesoscale circulation in the Ligurian basin.

**4.4 Transects 6: wind event E3**

The last transect T6 (December 15-18) is dominated by the wind event E3, that is mostly westerly but veering toward northerly during the last day. The results are summarized in Fig.6.

The hydrographic transects show a cooling of $\approx$ 1-1.5°C in the frontal region, together with a strong flattening of the isopycnals as shown also in Fig.7. The surface velocity depicted by the HF radar (Fig.6d) is mostly meridional and offshore,

in agreement with a wind response leading to upwelling. The geostrophic velocity is reduced (Fig.6c), and the westward jet structure appears fragmented. The corresponding zonal geostrophic transport is reduced with respect to the previous transects, and is even lower than for T3 and T4, reaching values of $\approx 0.45$ Sv (Fig.8). The comparison between surface geostrophic $u_{Sg}$ and radar based total velocity $\mathbf{u_S}$ (Fig. 9e) shows a significant deviation from geostrophy at the surface, as expected. $u_{Sg}$ has a complex pattern, while $\mathbf{u_S}$ is dominated by the meridional component ($\approx 30\%$ bigger than the zonal velocities).

Overall, the results are qualitatively similar to the ones of T2 and T3, describing an upwelling response to the wind that influences and weakens the geostrophic circulation. It is interesting to notice, though, that there are also some differences in the hydrographic properties with respect to the previous transects that are likely to be related to the transition from late-summer to winter conditions. The salinity minimum present in the first transect (Fig. 3b) vanishes with time and it is almost absent in T6 (Fig. 6b), suggesting that the late summer conditions in T1 are turning toward a more typical winter configuration, probably



due to the recurring effects of upwelling and mixing associated with the winter wind episodes. This is also shown by the pycnocline deepening of $\approx 30$ m in the offshore central region (Fig.7), occurring after the first two glider transects.

## 4.5 Summary of results

The main results from the above analysis can be summarized as follows.

- The observed variability of the Northern Current during the experiment period is dominated by wind response. In addition to this synoptic variability, the overall hydrographic conditions also suggest a seasonal trend, transitioning from late summer to fall-winter conditions.

- In absence of strong winds, the current is mostly zonal and geostrophic even at the sea surface. The associated zonal geostrophic transport over the water column is of the order of 1 Sverdrup.

- The response to strong westerly wind events (higher than $10 \ ms^{-1}$ and lasting more than $2-3$ days) induces offshore meridional currents at the surface and an upwelling response in the water column that flattens isopycnals in the frontal region. The westward zonal geostrophic current is weakened, and the associated zonal transport decreases, up to $40 - 50\%$, reaching values of $\approx 0.7 - 0.5$ Sv.

- A strong wind event lasting less than one day does not modify stratification and geostrophic velocity in the water column.

## 5 Direct surface response to the wind

In *Section 4* we have investigated the overall response of the NC to wind forcing, quantifying the variability of the water column stratification and of the zonal geostrophic transport. Here we focus on the processes that regulate sea surface currents response to direct wind forcing. Our final goal is to identify the ageostrophic wind induced surface velocity $\mathbf{u}_{SaW}$, and characterize it in terms of amplitude and angle with respect to the wind.

The difficulty lies in the fact that the quantity that is actually measured by HF radar is the total surface velocity $\mathbf{u_S}$, rather than $\mathbf{u}_{SaW}$. This is a general problem in the study of surface wind response. Information on $\mathbf{u_S}$ are provided by several instruments such as drifters, ADCP, or HF radar as in our case, and to decompose it in its various components (eq. 2) is not an easy task (Rio and Hernandez, 2003). This is especially true in coastal studies where altimeter based geostrophic velocities are not reliable. Often in experimental coastal studies, the surface velocity $\mathbf{u_S}$ is correlated with the wind to investigate which percentage of the current variance can be explained or studied at different scales, in order to decompose various processes (Kim et al., 2010).

Here, we follow a two step approach. As a first step, we consider the total velocity $\mathbf{u_S}$ measured by the HF radars, in order to obtain some general information on the surface response to the westerly wind events. In particular, we investigate the time evolution of the angle between the current and the wind. As a second step, we isolate $\mathbf{u}_{SaW}$ in selected periods, where we can provide an estimate of $\mathbf{u}_{Sg}$ based on the results of *Section 4*.





### 5.1 Investigation of the angle between wind and total surface velocity

Here we characterize the wind response in terms of the angle between the wind and the total surface currents $\mathbf{u_S}$ as measured by the radars. At each time interval of 1 h during the experiment period (Fig.2a), the angle $\alpha$ between the radar surface currents and the ALADIN winds is computed for all the available radar grid points, interpolating the winds over the radar grid. At each

time step, $\alpha$ is spatially averaged and time series of mean values and standard deviation *std* are generated.

Results for the mean angle $\bar{\alpha}(t)$ and *std(t)* are shown in Fig.10, with superimposed the time periods of the three wind events E1, E2, and E3 (Fig.2). Positive (negative) values indicate currents to the left (right) of the wind. During all wind events, $\bar{\alpha}$ is significantly negative, except for a short period in December 16, when a positive peak occurs. Notice though that the wind during that day dropped below the $10 \ ms^{-1}$ threshold (Fig.2b). When the wind is low, i.e. outside the event periods, the angle

oscillates between positive and negative values. Overall, the close response of the surface currents to the identified wind events provides a posteriori support to the choice of the $10 \ ms^{-1}$ wind speed threshold.

The results suggest a strong response of the total surface velocity to the wind events, with currents that tend to rotate to the right of the wind. The variability, quantified by the *std*, is high, even though mostly confined to negative values. This indicates the presence of some inhomogeneity in the wind response, as already evident in the radar averages of Figs.4,6. These

inhomogeneities can be due to many causes, from the configuration of the coast and/or the bathymetry (Kim et al., 2009), to the presence of mesoscale or submesoscale features, to inhomogeneities in the wind forcing over the radar coverage. Even though the ALADIN local wind is mostly homogeneous in the area of interest, wind gradients at larger spatial scales can also play a role in the surface currents response (Lebeaupin Brossier and Drobinski, 2009).

The average $\bar{\alpha}(t)$ values are $\approx -62.00°$ for E1, $\approx -93°$ for E2 and $\approx -56°$ for E3. Since the wind is prevalently westerly

(Fig.2b), this suggest that the current is prevalently moving offshore, in agreement with the results in *Section 4*. Possible reasons for the differences between the three events will be further discussed in *Section 5.3*. We notice that, of course the angle is not directly indicative of wind response since $\mathbf{u_S}$ contains also the geostrophic and ageostrophic residual components, and therefore $\bar{\alpha}$ cannot be compared with the angles predicted by the theoretical Ekman-like $\mathbf{u}_{SaW}$ solutions. In the following we will perform the decomposition of the geostrophic component from the HF radar total currents to estimate the magnitude and

angle of $\mathbf{u}_{SaW}$.

### 5.2 Estimation of the geostrophic component and ageostrophic wind response

Here we first of all assume that, during wind events the ageostrophic velocity is dominated by the wind induced component, i.e. $\mathbf{u}_{SaW} >> \mathbf{u}_{SR}$. This is partially justified by the fact that the $\mathbf{u}_{SR}$ processes are mostly high frequency, and oscillations from tides are typically small in the area (Albérola et al., 1995b; Arabelos et al., 2011) while inertial oscillations are expected

to be weaker during winter. Also, we recall that as discussed in *Section 3.5*, we obtain consistent results using raw and 36h low-pass filtered HF radar data. The geostrophic component $\mathbf{u}_{Sg}$, on the other hand, cannot be discarded since the results in *Section 4* show that $\mathbf{u}_{Sg}$ remains a sizable part of $\mathbf{u_S}$ in all cases (Fig.9), even though its transport is reduced in presence of westerly wind events (Fig.8).



As discussed above, performing the decomposition between ageostrophic and geostrophic component is challenging in most cases. In our case, we have information on the geostrophic velocity from the glider transects, but they are limited to the zonal direction and have restricted coverage in space and time. The results in *Section 4*, though, indicate that at least in some periods more extensive information can be obtained from the combination of glider and HF radar results.

During Transects 4-5, i.e. during the period December 9-15 when the wind was weak most of the time, the radar zonal velocity along the glider transect is very similar to the geostrophic one while the meridional velocity is reduced (Fig.9c,d). The corresponding radar velocity field over the whole region (Fig.5d) shows a well defined zonal current with weak meridional dependence. These results justify an ansatz that, during periods of weak winds, the geostrophic surface field $\mathbf{u}_{Sg}$ can be approximated on the basis of the HF radar velocity appropriately averaged or filtered. We also assume that this estimate,

indicated as $\widehat{\mathbf{u}_{Sg}}$, persists during wind episodes shorter than the time scale $T_W$, i.e. for episodes of the order of 1 day. This is in agreement with the results in *Section 4* regarding E2 (lasting less than 1 day) that suggest that the E2 winds do not significantly influence the stratification and the geostrophic velocity.

This ansatz is used to perform the decomposition and to study the wind response during E2. The geostrophic velocity $\widehat{\mathbf{u}_{Sg}}$ is estimated averaging the radar velocity over a time period $T$ prior to the onset of the wind event, and the ageostrophic wind

response $\mathbf{u}_{SaW}$ is estimated subtracting $\widehat{\mathbf{u}_{Sg}}$ from the total radar velocity $\mathbf{u}_S$. A sensitivity study is carried out varying $T$ in the range of 6-12 hours, and the rms difference between the results is $\approx 20\%$ for $\bar{\alpha}(t)$.

An example of the geostrophic decomposition for a selected surface current field during E2, on December 12 at 14h, is shown in Fig.11. The estimated $\widehat{\mathbf{u}_{Sg}}$ field, considering the basic case of $T$=6 h for surface currents average, and the ALADIN wind are shown in Fig.11a,c. The HF radar total velocity $\mathbf{u}_S$ is shown in Fig.11b, while the estimated $\mathbf{u}_{SaW}$, obtained subtracting

the field in Fig.11a from the field in Fig.11b, is shown in Fig.11d. Superimposed colors refer to the angle $\alpha$ of surface currents $\mathbf{u}_{SaW}$ with respect to the wind. Subtracting the zonal westward geostrophic component to the total current field basically correspond to rotating currents eastward and it results in decreasing the angle with respect to the westerly winds. The angle is negative for most of the field, except for a few grid points in the north-eastern corner of the radar field.

Time series of spatially averaged results are shown in Fig.12, for periods within E2 (left panels) and E3 (right panels) wind

events. The estimated $\widehat{\mathbf{u}_{SaW}}$ is characterized in terms of magnitude and angle with respect to the wind (Fig.12a) averaged over all the radar grid points. Also, the percentage of negative angle values distributed over the radar field is shown in Fig.12c, as an additional measure of variability. The angle between the wind and the ageostrophic component has a time average value $\overline{\alpha(t)}$ $\approx -28.00°$, while the magnitude of the ageostrophic component reaches values of $\approx 25 cms^{-1}$, comparable to the average magnitude of the geostrophic component and corresponding to the $\approx 2\%$ of the ALADIN wind average magnitude (about

$13 ms^{-1}$). Negative angles, i.e. ageostrophic component to the right of the wind, prevail over the radar current field (up to about 80% of all grid points) during the wind event (Fig.12c).

The same method has also been applied to the onset of the wind event E3, considering only the first day of the wind episode, when we can assume that the estimate of the geostrophic velocity holds. Results in Fig.12b show the angle between the wind and the ageostrophic component, with time average $\overline{\alpha(t)} \approx -26°$, while the ageostrophic component magnitude reaches values

of $\approx 30 cms^{-1}$, comparable to the average magnitude of the geostrophic component and corresponding to the $\approx 2.5\%$ of the





ALADIN wind average magnitude (about $12 \text{ms}^{-1}$). Negative angles, i.e. ageostrophic component to the right of the wind, prevail over the radar current field (up to about 90% of all grid points) during the wind event (Fig.12d).

The surface current response to both wind events considered here shows similarities in term of the angle between the ageostrophic component and the wind direction (about -25.00°) and also considering the magnitude of the ageostrophic component compared to wind speed (about 2% as previously observed by Chang et al. (2012) and Poulain et al. (2009)). Surface currents appear to respond to the wind quite homogeneously in space over the whole HF radar field.

### 5.3 Discussion

The results in *Sections 5.1-5.2* show that the average angle between the surface current and the wind is very different for the total surface current $\mathbf{u_S}$ ($\bar{\alpha} \approx -55°$ to $-90°$) and for the wind driven ageostrophic component $\mathbf{u}_{SaW}$, ($\bar{\alpha} \approx -25°$ to $-30°$). This highlights the importance of subtracting the geostrophic component of the velocity, especially in a boundary current situation where it is very relevant. The correction decreases the angle to the right of the wind, as it can be expected since the geostrophic velocity is primarily zonal and westward while the wind is mostly westerly.

More in details, the different values of $\bar{\alpha}$ for $\mathbf{u_S}$ during the three wind events (Fig.10) can be due to a number of reasons. A first hypothesis is that they are linked to different values of the geostrophic velocity $\mathbf{u}_{Sg}$. From the results in *Section 4*, $\mathbf{u}_{Sg}$ is expected to be stronger and more zonal during E2 and at the beginning of E1 and E3, i.e. when the wind has not yet acted to weaken it. This could explain the observed values of $\bar{\alpha} \approx -90°$ during those periods. As the time progresses during the wind events E1 and E3, the zonal geostrophic velocity weakens and as a consequence the angle is expected to decrease, as shown in Fig.10. An other possible reason for the variability in $\bar{\alpha}$ values is the presence of time varying inhomogeneities in the field, as suggested by the current reversals in Fig.4, that could be due for instance to the interaction with the outflow from the Gulf of Lyon (Schaeffer et al., 2011).

For $\mathbf{u}_{SaW}$, the values of $\bar{\alpha}$ are more similar in the two cases considered and the variability is reduced. It is interesting to compare these results with previous results in the literature, even though the comparison is challenging due to the use of different data and methods. A number of recent results are based on subtracting the geostrophic component estimated from altimetry data, and they consistently show angles to the right of the wind (in the northern hemisphere). Results from HF radar in the Kuroshio area (Tokeshi et al., 2007) suggest values of $\approx 38 - 48°$, while results from SVP drifters with drogue at 15 m (Rio and Hernandez, 2003) provide values of $\approx 10 - 40°$ at global scales for latitudes higher than $30°N$. In the Black Sea, SVP drifter data (Stanichny et al., 2016) suggest values of $\approx 13°$ at the sea surface. Finally in the Mediterranean Sea (Poulain et al., 2009) find from SVP and CODE drifters (drogued at 1 m) values of $\approx 27 - 42°$ and $\approx 17 - 20°$ respectively, obtained without subtracting the geostrophic component. Overall, these values suggest a range of $\approx 10 - 40°$, that is consistent with our results. With respect to our results, though, we notice that most of the previous results have been obtained considering a larger scale geostrophic component and longer time scales of a few days. An exception is given by the work of Sentchev et al. (2017) in the Toulon area, that considers daily wind oscillations corresponding to light sea breeze. Results from HF radar and ADCP in this case suggest angles of $\approx 15 - 20°$ to the left of the wind, indicating a different balance with respect to the typical Ekman balance.





An important final remark is the fact that, as pointed out in *Section 3*, estimates of $\mathbf{u}_{SaW}$ based on HF radars and surface drifters are likely to be at least partially biased by the Stokes drift-like component of the velocity (Ardhuin et al., 2009). This component is expected to be in the same direction as the wind, therefore causing a bias that tends to decrease the value of the estimated angle. In our case, then, the angle of the actual Eulerian velocity could exceed $-30°$. This issue, that is common to all the works based on HF radar and surface drifters, is outside the scope of the present paper but it will be considered in future works, considering additional wave spectra information.

## 6 Summary and conclusions

In this paper, a multi-platform observing system is used to monitor the variability of the boundary current of the North-Western Mediterranean Sea, i.e. the Northern Current. The adopted multi-platform system gives synoptic measurements of currents and water masses properties at spatio-temporal scales that cannot be resolved only with classical vessel surveys or satellite remote sensing. We use water column data from repeated glider transects and vessel surveys, surface current fields from HF radar, wind time series from a meteo station and an atmospheric model to describe the evolution of the NC off Toulon for a period of approximately two weeks in December 2011.

The NC variability is dominated by a synoptic response to wind events, even though a seasonal trend is also observed, transitioning from late summer to fall-winter conditions. When the wind is weak, the current is mostly zonal and in geostrophic balance even at the surface, with a zonal transport associated to the NC of $\approx 1$ Sv. During two strong westerly wind events lasting longer than 2-3 days, an upwelling response is observed, with offshore surface transport, surface cooling, flattening of the isopycnals and reduced zonal geostrophic transport (0.5-0.7 Sv). When the wind lasts less than one day, surface currents respond to winds but the water column stratification and the geostrophic transport are not affected because the wind event is not persistent enough.

We also specifically investigate the surface currents response to the wind. The total surface current as observed by the HF radar is found to respond to the wind events, rotating at $\approx -55°$ to $-90°$ to the right of the wind. During the first day of selected wind events, we also perform a decomposition between geostrophic and ageostrophic components of the surface current using results from glider and HF radar. The directly wind driven ageostrophic component is found to rotate of a smaller angle $\approx -25°$ to $-30°$ to the right of the wind. The ageostrophic component magnitude corresponds to $\approx 2\%$ of the wind speed.

This paper provides a first step in the joint use of glider and HF radar data to describe the variability of a boundary current in terms of both geostrophic and ageostrophic processes. Results show a high synoptic variability of the geostrophic component related to wind episodes persistent enough to modify water column stratification and pressure gradients, pointing out to the difficulties of decomposing flow dynamics according to time scales and forcings (Kim et al., 2010).

The decomposition in geostrophic and ageostrophic velocity was carried out for space and time scales smaller than in most previous works (Rio and Hernandez, 2003; Tokeshi et al., 2007), i.e. of the order of 1 day and tens of km, as appropriate scales for the Northern Current. Further developments in the decomposition method are foreseen using time dependent geostrophic





velocities. This approach will be first tested using models results with an OSSE type of approach, and we expect that it will provide useful insights for data assimilation and data blending applications.

A number of interesting issues, that are not considered here will be considered in future works. They include nonlinear wind response in the frontal area, and interactions with mesoscale and submesoscale instabilities that can modulate the geostrophic
5   and ageostrophic response. Also, the effects of the bias due to the Stokes-like term in the HF radar velocity retrievals need further investigation.

More generally, following this and other specific applications based on the combination of autonomous observing platforms with classical marine surveys, several other multi-platform observing systems are recently developing as a transnational effort within the Mediterranean oceanographic community. These systems are making available unprecedented synoptic high-
10   resolution datasets that could be useful not only for specific scientific purposes but also for practical applications such as the management of coastal and marine shared resources.

*Acknowledgements.* The analysis of the dataset has been supported and co-financed by the JERICO-NEXT project. This project has received funding from the European Union's Horizon 2020 research and innovation programme under grant agreement No. 654410. The multi-platform experiment has been carried out within the TOSCA project, co-funded by the European Regional Development Fund in the
15   framework of the MED program. Wind data were kindly made available by MeteoFrance. The authors wish to thank the Urania R/V's crew that made the experiment possible, the glider support team of DT-INSU, and the MIO's HF radar team.



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



| | *glider* | |
| --- | --- | --- |
| | **Start** | **End** |
| **Transect 1** | 02 Dec. 11:57 | 04-Dec 20:15 |
| **Transect 2** | 04 Dec. 21:50 | 07-Dec 09:32 |
| **Transect 3** | 07 Dec. 11:17 | 09-Dec 13:38 |
| **Transect 4** | 09 Dec. 15:21 | 12-Dec 21:16 |
| **Transect 5** | 12 Dec. 21:45 | 15-Dec 13:15 |
| **Transect 6** | 15 Dec. 15:09 | 18-Dec 11:34 |

**Table 1.** Dates of glider transects.




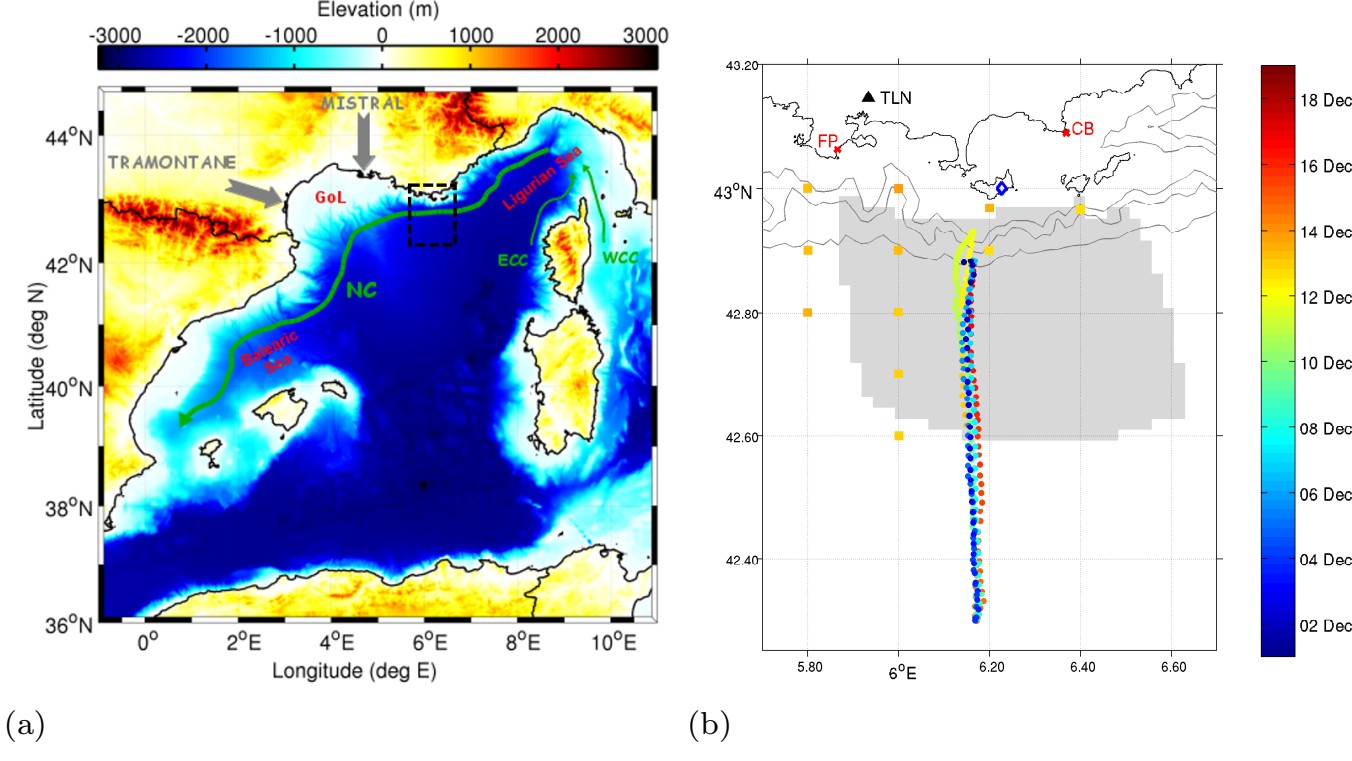

**Figure 1.** (a) The Western Mediterranean Sea, including northern circulation branches and main winds. The region of study is marked as a dashed black square. The following acronyms are used: GoL: Gulf of Lion, NC: Northern Current, WCC: Western Corsica Current, ECC: Eastern Corsica Current. (b) Details of the region of study including the synoptic measurements carried out during the experiment (December 2-19, 2011). The gray area is the HF radar surface current field coverage, the yellow squared marks represent the CTD stations, while the dots array at 6.20° E is the track of the repeated glider transects. CTD stations and glider tracks are color coded in time. The blue diamond indicates Porquerolles wind station's position. Bathymetric lines at 500 m, 1000 m and 2000 m depth. The following acronyms are used: TLN: Toulon, FP: Fort Peyras, CB: Cap Bénat.




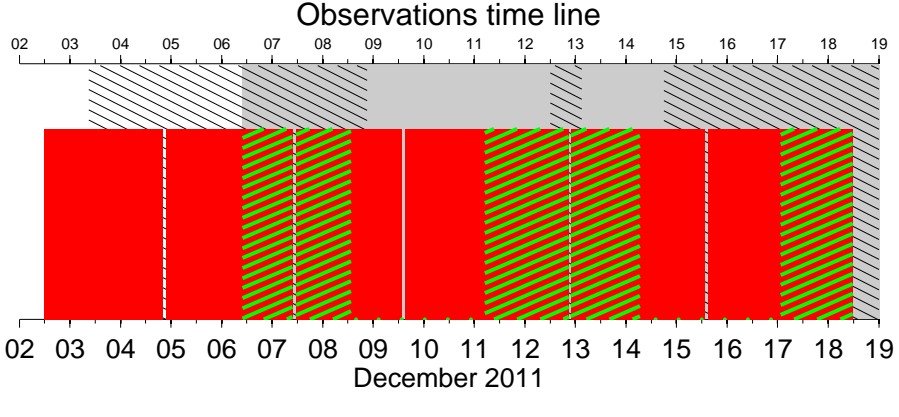

(a)

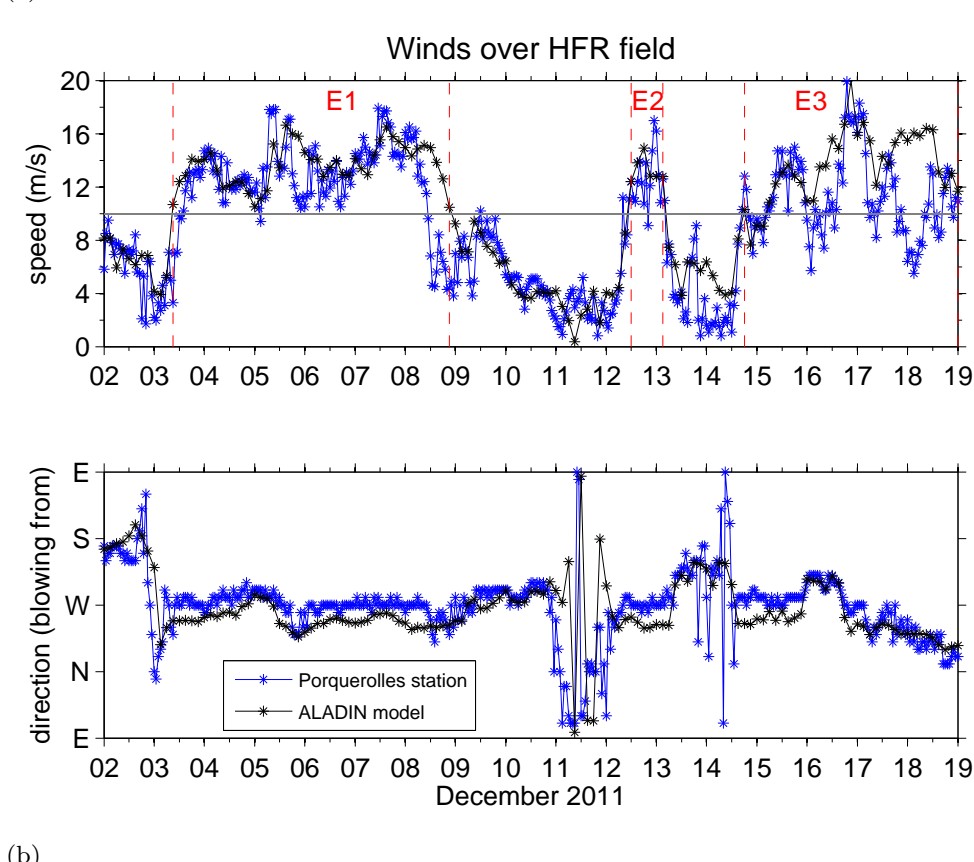

(b)

**Figure 2.** (a) The multi-platform observations time line: the black dashed lines indicate the selected wind events (compare with panel below), the gray area indicates the radar dataset availability, the red areas indicate the glider transects, and the green dashed lines indicate the period in which the glider transects fall inside the HFR coverage. (b) Wind time series from Porquerolles wind station (blue line) and from ALADIN model (black line) are compared in terms of speed (middle panel) and direction (bottom panel). The limits of the selected wind events (speed>10ms$^{-1}$, gray line in the middle panel) are indicated with red dashed lines and named E1, E2, and E3.



**Figure 3.** Potential temperature (a), salinity (b), and geostrophic velocity (c) for glider Transect 1 (see also Table 1 for time period definition). Potential density anomaly isolines (black) in panels a and b.



**Figure 4.** Potential temperature (a), salinity (b), and geostrophic velocity (c) for glider Transect 3 (see also Table 1 for time period definition). Potential density anomaly isolines (black) in panels a and b. The black dashed line (panel c) represents the southern boundary of the HF radar field. In panel (d), black arrows are the surface currents from HFR averaged during the period in which the glider transect (green dashed line) falls inside the HFR coverage (and considering only grid points with at least 80% available data over the time period). The blue arrow represent the wind from Porquerolles station averaged over the same period considered for the HFR average.



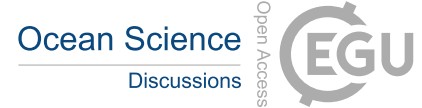
**Figure 5.** Potential temperature (a), salinity (b), and geostrophic velocity (c) for glider Transect 5 (see also Table 1 for time period definition). Potential density anomaly isolines (black) in panels a and b. The black dashed line (panel c) represents the southern boundary of the HF radar field. In panel (d), black arrows are the surface currents from HFR averaged during the period in which the glider transect (green dashed line) falls inside the HFR coverage (and considering only grid points with at least 80% available data over the time period). The blue arrow represent the wind from Porquerolles station averaged over the same period considered for the HFR average.



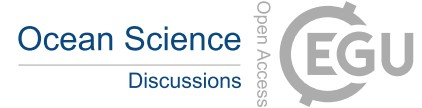

**Figure 6.** Potential temperature (a), salinity (b), and geostrophic velocity (c) for glider Transect 6 (see also Table 1 for time period definition). Potential density anomaly isolines (black) in panels a and b. The black dashed line (panel c) represents the southern boundary of the HF radar field. In panel (d), black arrows are the surface currents from HFR averaged during the period in which the glider transect (green dashed line) falls inside the HFR coverage (and considering only grid points with at least 80% available data over the time period). The blue arrow represent the wind from Porquerolles station averaged over the same period considered for the HFR average.





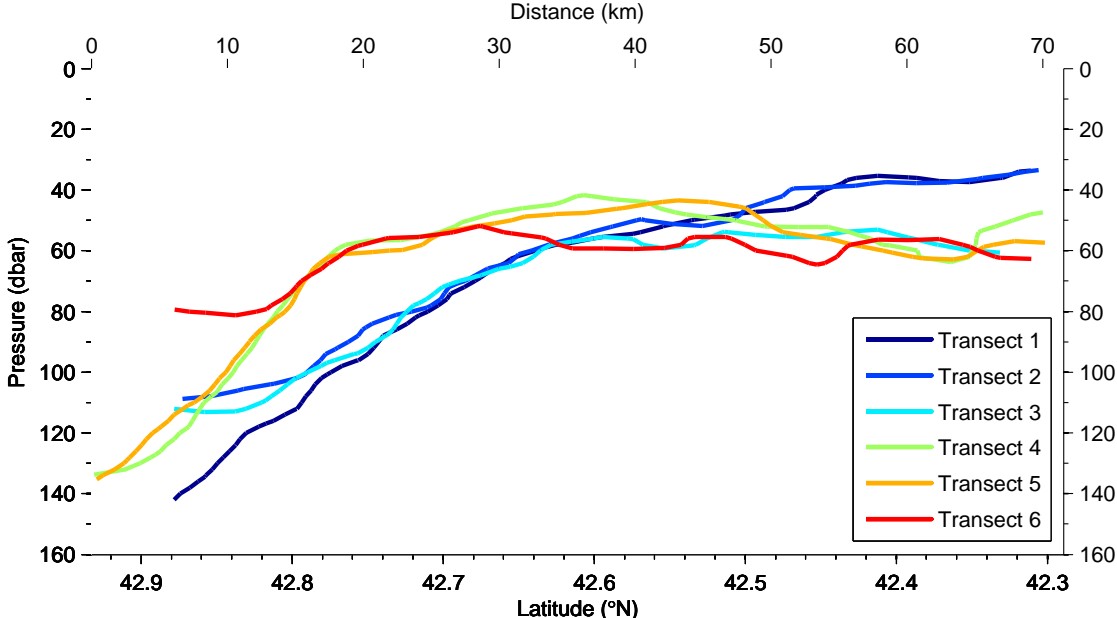

**Figure 7.** Potential density anomaly $\sigma(z)$=28.7 kg/m$^3$ isopycnal evolution color coded for each glider transect.



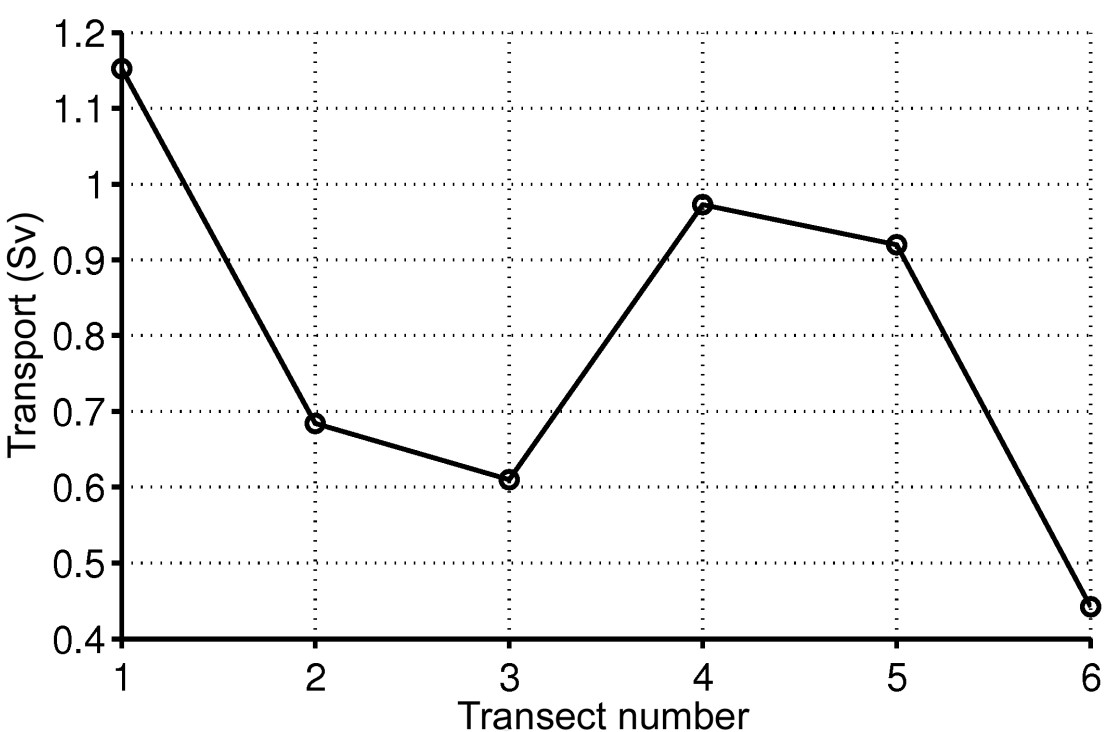

**Figure 8.** Cross-shore NC transport computed from glider relative geostrophic velocity profiles (reference pressure at 500 mbar) for each glider transect.





**Figure 9.** Comparison of $u_S$ (solid black line) and $v_S$ (dashed black line) components of total surface currents from HFR and $u_{Sg}$ (green line) geostrophic component from glider transects. Positive u (v) currents are eastward (northward), and the gray line indicates null velocities. Panels (a) to (e) represent measurements during Transects 2 to 6.




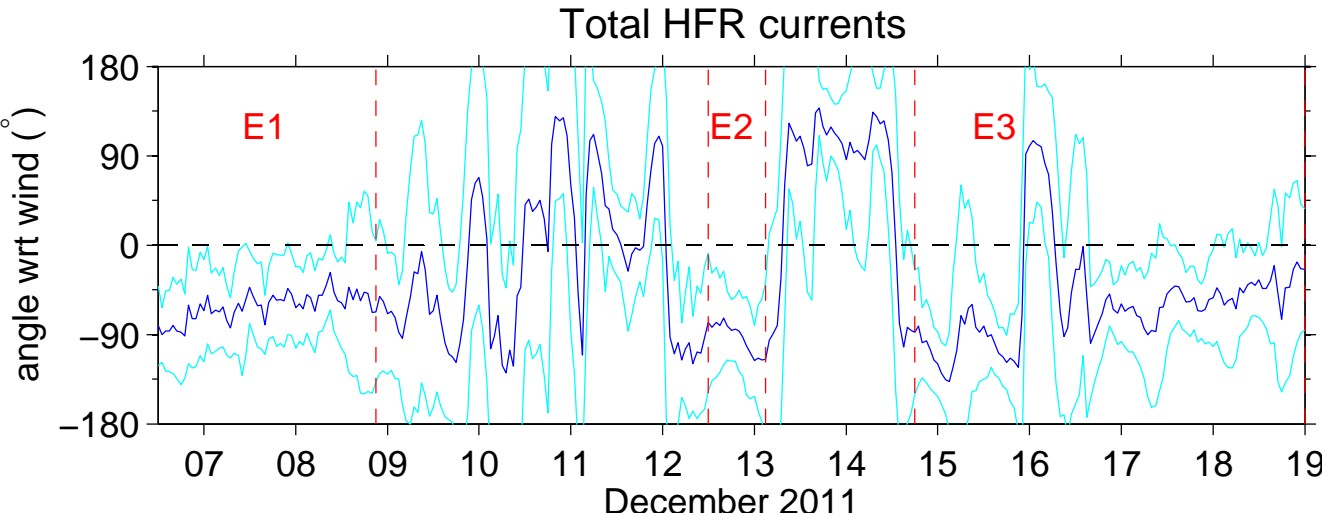

**Figure 10.** Angle $\bar{\alpha}(t)$ between total surface currents $\mathbf{u_S}$ and average ALADIN winds over the HFR coverage (blue line). Negative angles indicate currents to the right of the wind direction. The cyan lines represent the std. The limits of the selected wind events (E1, E2, and E3) are indicated with red dashed lines (see Fig. 2b for more details).





**Figure 11.** Example maps of: (a) 6h time-averaged geostrophic flow $\mathbf{u}_{Sg}$ derived from HFR, (b) instaneous HFR surface currents $\mathbf{u_S}$, (c) ALADIN wind field, and (d) ageostrophic surface currents $\mathbf{u}_{SaW}$ superimposed on the color coded map quantifying the angle $\alpha$ between the ageostrophic component and ALADIN wind. Negative values means current to the right of the wind.





**Figure 12.** (a,b) Time series of the spatially averaged ageostrophic component $\widehat{\mathbf{u}_{SaW}}$ magnitude (top panel) and angle $\alpha(t)$ (bottom panel) with respect to the wind direction (negative angle means currents to the right of the wind). The cyan lines in the bottom panels represent the std. The black dashed line represents the average magnitude of the geostrophic component $\widehat{\mathbf{u}_{Sg}}$. (c,d) Time series of the percentage of negative angles $\alpha$ (between ageostrophic component and wind) distributed over the HFR coverage. The black dashed line indicates 50%.