# Peer review of "Wind induced variability in the Northern Current (North-Western Mediterranean Sea) as depicted by a multi-platform observing system"

_Ocean Science, 2018_

## Referee Comment (RC1) · Anonymous Referee #1 · 19 Apr 2018

Review on

Wind induced variability in the Northern Current (North-Western Mediterranean Sea) as depicyed by a multi-platform observing system

by

Maristella Berta, Lucio Bellomo, Annalisa Griffa, Marcello Magaldi, Anne Molcard, Carlo Mantovani, Gian Pietro Gasparini, Julien Marmain, Anna Vetrano, Laurent Beguery, Mireno Borghini, Yves Barbin, Joel Gaggelli, and Celine Quentin

[Figure]

General Comments: =================

This paper discusses the variability of the Northern Current south of Toulon based on measurements by a glider, ship borne CTD, HF radar, as well as wind speed and direction based on observations and from an atmospheric model. The authors try to split the surface current as measured by HF radar into a wind-driven and a geostrophic component. The ship borne CTD measurements (11 stations during 5 days) have only been used to calibrate the glider data (6 transects during 16 days). HF radar surface currents are avaiable during 17 days. This time frame is quite restricted which makes the results shown in the manuscript interesting but not generally valid in a statistic sense.

The procedure to derive the wind driven current component is quite simple. In a first step, the angle between model winds and HF radar surface currents is determined and the mean value and its standard deviation is calculated. If the surface current is dominated by wind, this angle should be negative, i.e. to the right of the wind following Ekman's theory, while during weak wind, the HF radar surface currents mainly represent the geostrophic component. The geostrophic component measured prior to the onset of a wind event should also represent the geostrophic component during the wind event, as only longer lasting events of strong wind mix the water masses and modify the geostrophic component.

There is another method to find the angle and the wind driven component of the current directly by using a complex correlation (Kundu 1976) between wind and surface current, see Essen 1993, Section 2.1 on the Ekman component. It would be interesting, if the complex correlation gives comparable results. Remarks on the Stokes drift component (page 8/line 14 of this manuscript) can be found in Section 2.2 of Essen's paper.

Also, there is another quite old paper by Essen 1995 on the derival of the geostrophic current from HF radar surface currents and SST images. This paper might give some

ideas on how to join HF radar surface currents and glider zonal currents to derive the geostrophic component of the current.

The manuscript is well structured and written in good English grammer. There are only a few detailed comments and remarks on typos, see below.

Papers that should be discussed:

(1) Essen, H.-H., Ekman Portion of Surface Currents, as Measured by Radar in Different Areas. Deutsche Hydrographische Zeitschrift 45, pp. 57...85, 1993.

(2) Essen, H.-H., Geostrophic surface currents as derived from satellite SST images and measured by a land-based HF radar. Int. Journ. Rem. Sens. 16, pp. 239...256, 1995.

Detailed comments (page/line numbers): ====================================

3/35: "the use of altimeters is not appropiate." Please add a remark on why this is the case.

7/23: Please deine "GDOP".

9/02: "is consistant with the Beaufort scale..." Please explain. 5 Beaufort is 8.0 - 10.7 m/s, 6 Beaufort is 10.8 - 13.8 m/s. What is consistant here?

14/06: What is the location of the mean angle shown in Fig. 10?

14/23: Here the complex correlation between wind and current might give more insight.

17/15: "transitioning from late summer to fall-winter conditions." You have a bit more than two weeks of data only. Is this really enough to identify a seasonal trend?

18/01: Please define "OSSE".

Typos (page/line numbers): ==========================

5/33: "(x.y.t)" -> "(x,y,t)" 5/33: "to be subtracted to" -> "to be subtracted from"

6/30 - 7/1: check '( ... )'

7/33: "16.1 MHz,the" -> "16.1 MHz, the"

10/29: "are colder of ∼1.5..." -> "are colder by ∼1.5..."

10/30: "part is deepened of ∼20-30..." -> part is deepened to ∼20-30..."

10/33: "it deepenes of ∼20-30..." -> "it deepenes to ∼20-30..."

11/19: "transport reduced of ∼50%..." -> "transport reduced by ∼50%..." ????

12/01: "geostrophic velocity, structure..." -> geostrophic velocity structure..." 12/01: "it influences surface velocity." -> "it influences the surface velocity."

Figures: ========

Fig 1b: "the yellow squared marks represent the CTD stations" They are orange due to the color coding with time. Also, please mention the HF radar sites at FP and CB in the caption.

Fig 3c: Please add a remark on the black dashed line in the caption.

Fig.4d: "black arrows are the surface currents" -> "black arrows show the surface currents", same in Fig. 5d, 6d.

---

## Referee Comment (RC2) · Anonymous Referee #2 · 23 May 2018

This paper deals with surface current observation by HF radar in the framework of a coastal slope current (The Northern Current, front of Toulon, N-W Mediterranean Sea). The effect of a wind blowing in opposite direction is investigated. Thanks to a repeated cross-current glider line a geostrophic evaluation of the current is derived from the temperature/salinity transects. Authors propose an estimation of the wind induced surface current subtracting to a measured surface current an underlying geostrophic current.

This paper is well written and very concise; the authors prevent systematically objec-

tions, showing they are aware on the potential limitations or difficulties of this approach : - Inertial oscillation remain "weak" in winter (page 14) - The fetch is to short to take into account the Stokes drifts (page 8) but should be taken into account for future work (page 18). - Mesoscale structures are included in geostrophic part when sub-mesoscale features are put into a residual unresolved part of the current.

One of the interesting observation is the offshore spreading of the Atlantic Waters, flattening the isopycnals and decreasing the westwards zonal transport. To my mind, the generation of mesoscale structures south of the main vein of current (observed in geostrophic currents on figure 5/6/7 ) is underexploited and under-commented. Despite theses structures are out of the radar spot, it should be discussed. Then the paper title would be more in agreement with it contents.

The main weakness of this paper lies in the purely kinematic point of view of the demonstration regarding the velocity decomposition. We must agree with the authors. Despite the limited reach of the paper, data proposed are original and interesting. I suggest, this paper remains suitable for publication after minor modifications.

Detailed comments :

Page 6/7 chapter 3.3. The CTD's performed by the vessel survey, could be probably used to asses the level of no motion in the thermal wind approximation.

Page 9 line 2 : The reference to the "Beaufort scale" seems to be not relevant in this context.

Page 10 line 15 : The reference to the Guibout "Atlas" is not sufficient. Please describe shortly the "end of summer conditions" and interpret the observed salinity minimum (present but not commented in Guibout).

Page 10 line 20 : date(1995) or indices(a) are missing in the references for this two citations. (Alberola, Petrenko)

Page 32 figure 8 : really cross-shore ? : along shore , cross section or zonal transport

would be more appropriate.

---

## Author Comment (AC1) · 15 Jun 2018

Our comments to both reviewers and the marked version of the modified manuscript were uploaded in the form of a supplement zip file.

Please also note the supplement to this comment:
https://www.ocean-sci-discuss.net/os-2018-20/os-2018-20-AC1-supplement.zip

---

## Author Response (AR1)

Dear Editor,

Enclosed herewith please find the revision of the manuscript entitled: "Wind induced variability in the Northern Current (North-Western Mediterranean Sea) as depicted by a multi-platform observing system", referenced as os-2018-20.

We thank the reviewers for their thorough reading of the text and useful comments. Below is our detailed response and description of the modifications in the manuscript, while the attached new version of the paper explicitly shows all revisions in red.

We hope that the manuscript is now satisfactory for publication in Ocean Science.

=======================
Anonymous Referee #1
========

- *General comments:*

  *This paper discusses the variability of the Northern Current south of Toulon based on measurements by a glider, ship borne CTD, HF radar, as well as wind speed and direction based on observations and from an atmospheric model. The authors try to split the surface current as measured by HF radar into a wind-driven and a geostrophic component. The ship borne CTD measurements (11 stations during 5 days) have only been used to calibrate the glider data (6 transects during 16 days). HF radar surface currents are available during 17 days. This time frame is quite restricted which makes the results shown in the manuscript interesting but not generally valid in a statistic sense.*

  *The procedure to derive the wind driven current component is quite simple. In a first step, the angle between model winds and HF radar surface currents is determined and the mean value and its standard deviation is calculated. If the surface current is dominated by wind, this angle should be negative, i.e. to the right of the wind following Ekmans theory, while during weak wind, the HF radar surface currents mainly represent the geostrophic component. The geostrophic component measured prior to the onset of a wind event should also represent the geostrophic component during the wind event, as only longer lasting events of strong wind mix the water masses and modify the geostrophic component.*

  *There is another method to find the angle and the wind driven component of the current directly by using a complex correlation (Kundu 1976) between wind and surface current, see Essen 1993, Section 2.1 on the Ekman component. It would be interesting, if the complex correlation gives comparable results. Remarks on the Stokes drift component (page 8/line 14 of this manuscript) can be found in Section 2.2 of Essens paper.*

  R: We thank the reviewer for the suggestions. We have computed the complex correlation between wind and ocean currents with the method introduced by Kundu (1976) and the results (shown in Fig.1 at the end of our response) are now discussed in the text on page 15 lines 5-13. The phase angle is comparable, within the std range, to the time series in Fig.10 of the manuscript. The average correlation magnitude between total currents and winds has values around 0.4 for the three wind events, given that total currents measured by HFR include the Ekman response to winds as well as other surface processes not directly wind related. See also our response to detailed comment 14/23 (page 3 of this file).

We have also added the reference for the Stokes drift on page 8 line 20 of the new manuscript.

*Also, there is another quite old paper by Essen 1995 on the derival of the geostrophic current from HF radar surface currents and SST images. This paper might give some ideas on how to join HF radar surface currents and glider zonal currents to derive the geostrophic component of the current.*

R: We have added a comment on this suggestion in the Conclusion (page 18 lines 27-28).

*The manuscript is well structured and written in good English grammar. There are only a few detailed comments and remarks on typos, see below.*

*Papers that should be discussed:*

*(1) Essen, H.-H., Ekman Portion of Surface Currents, as Measured by Radar in Different Areas. Deutsche Hydrographische Zeitschrift 45, pp. 57...85, 1993.*

*(2) Essen, H.-H., Geostrophic surface currents as derived from satellite SST images and measured by a land-based HF radar. Int. Journ. Rem. Sens. 16, pp. 239...256, 1995.*

R: we have discussed these references on page 8 line 20 and page 18 lines 27-28.

- *Detailed comments (page/line numbers):*

*3/35: "the use of altimeters is not appropriate." Please add a remark on why this is the case.*

R: in the new version of the manuscript (pages 3;4 lines 35;1-4) we have now specified that for coastal boundary currents such as the NC and for the space and time scales we are interested in (tens of kilometers and days) the use of satellite altimetry requires special attention (Vignudelli et al., 2000; Gómez-Enri et al., 2016) and is often not appropriate because of problems of accuracy, land contamination and low space and time resolution at least for global products such as AVISO (Bouffard et al., 2008; Berta et al., 2015).

*7/23: Please define "GDOP".*

R: we have expanded the acronym GDOP (Geometric Dilution Of Precision) on page 7 line 29.

*9/02: "is consistent with the Beaufort scale..." Please explain. 5 Beaufort is 8.0 - 10.7 m/s, 6 Beaufort is 10.8 - 13.8 m/s. What is consistent here?*

R: we have rephrased the point (page 9 lines 9-11) clarifying that the chosen wind speed threshold ($10$ ms$^{-1}$) is consistent with previous Mistral observations in the Toulon area, such as Caccia et al. (2004) and Guénard et al. (2005).

*14/06: What is the location of the mean angle shown in Fig. 10?*

R: we have further clarified (page 14 line 22) that results shown in Fig.10 represent the angle $\bar{\alpha}(t)$ averaged over the HFR field at each time step. Also, we have further specified

in Fig.10 caption that $\bar{\alpha}(t)$ is spatially-averaged.

***14/23: Here the complex correlation between wind and current might give more insight.***

R: we thank the reviewer for this suggestion. We have applied the complex correlation method (Kundu, 1976) to our dataset to estimate the magnitude of the correlation between wind and total currents, as well as the phase angle between wind and currents. The phase angle estimated with this alternative method is comparable, within the std range, to the time series in Fig.10 of the manuscript. The average correlation magnitude between total currents and winds has values around 0.4 for the three wind events, given that total currents measured by HFR include the Ekman response to winds as well as other surface processes not directly wind related. The new analysis is shown in Fig.1 at the end of our response file, and the new results are discussed on page 15 lines 5-13 of the new manuscript.

***17/15: "transitioning from late summer to fall-winter conditions." You have a bit more than two weeks of data only. Is this really enough to identify a seasonal trend?***

R: we agree that the expression "seasonal trend" does not fit with the time window of our dataset. We have rephrased the expression (abstract lines 3-4, page 13 lines 21-23 and page 18 lines 5-7) specifying that the hydrographic transects display flattening of isopycnals and deepening of the mixed layer offshore, evidencing early stages of the transition from late summer to fall-winter typical conditions as observed by Guibout (1987); Albérola et al. (1995)

***18/01: Please define "OSSE".***

R: we have expanded the acronym OSSE (Observing System Simulation Experiment) on page 18 line 25.

- ***Typos (page/line numbers):***

***5/33: "(x.y.t)" → "(x,y,t)" 5/33: "to be subtracted to" → "to be subtracted from"***

R: we have corrected the typos on page 6 line 3 of the new manuscript.

***6/30 - 7/1: check "( … )"***

R: we have removed the extra brackets on page 7 lines 5 and 7.

***7/33: "16.1 MHz,the" → "16.1 MHz, the"***

R: we have removed the typo on page 8 line 7.

***10/29: "are colder of $\sim$ 1.5..." → "are colder by $\sim$ 1.5..."***

R: we have removed the typo on page 11 line 7.

***10/30: "part is deepened of $\sim$20-30..." → "part is deepened to $\sim$20-30..."***

R: we have removed the typo on page 11 line 8.

*10/33: "it deepens of ∼20-30..." → "it deepens to ∼20-30..."*

R: we have removed the typo on page 11 line 11.

*11/19: "transport reduced of ∼50%..." → "transport reduced by ∼50%..." ????*

R: we have removed the typo on page 12 line 1.

*12/01: "geostrophic velocity, structure..." → "geostrophic velocity structure..."*
*12/01: "it influences surface velocity." → "it influences the surface velocity."*

R: we have removed the typos on page 12 line 16.

- *Figures:*

*Fig 1b: "the yellow squared marks represent the CTD stations" They are orange due to the color coding with time. Also, please mention the HF radar sites at FP and CB in the caption.*

R: we have rephrased the caption as suggested.

*Fig 3c: Please add a remark on the black dashed line in the caption.*

R: we have added the description of the black dashed line.

*Fig.4d: "black arrows are the surface currents" → "black arrows show the surface currents", same in Fig. 5d, 6d.*

R: we have corrected all captions as suggested.

========================
Anonymous Referee #2

========

- *General comments:*

  *This paper deals with surface current observation by HF radar in the framework of a coastal slope current (The Northern Current, front of Toulon, N-W Mediterranean Sea). The effect of a wind blowing in opposite direction is investigated. Thanks to a repeated cross-current glider line a geostrophic evaluation of the current is derived from the temperature/salinity transects. Authors propose an estimation of the wind induced surface current subtracting to a measured surface current an underlying geostrophic current.*

  *This paper is well written and very concise; the authors prevent systematically objections, showing they are aware on the potential limitations or difficulties of this approach: - Inertial oscillation remain "weak" in winter (page 14) - The fetch is to short to take into account the Stokes drifts (page 8) but should be taken into account for future work (page 18). - Mesoscale structures are included in geostrophic part when submesoscale features are put into a residual unresolved part of the current.*

  *One of the interesting observation is the offshore spreading of the Atlantic Waters, flattening the isopycnals and decreasing the westwards zonal transport. To my mind, the generation of mesoscale structures south of the main vein of current (observed in geostrophic currents on figure 5/6/7 ) is underexploited and under-commented. Despite theses structures are out of the radar spot, it should be discussed. Then the paper title would be more in agreement with it contents.*

  *The main weakness of this paper lies in the purely kinematic point of view of the demonstration regarding the velocity decomposition. We must agree with the authors. Despite the limited reach of the paper, data proposed are original and interesting. I suggest, this paper remains suitable for publication after minor modifications.*

  R: we thank the reviewer for the fruitful comments, we have further discussed the "generation mesoscale structures" on page 11 lines 22-26 and page 13 lines 6-7. In addition to the upwelling, other phenomena related to wind response are likely to occur due to mixing that deepens the thermocline and possibly to convection processes. Within the first 50-60 m depth we also observe recirculating cells further offshore the NC front associated to the wind-driven modification of the water masses circulation offshore Toulon. This behavior is in analogy with NC variability and meanders observations off Nice in autumn-winter by Albérola et al. (1995); Sammari et al. (1995). The recirculating cells are not visible in the surface current map because they lay outside the HFR field.

- *Detailed comments:*

  *Page 6/7 chapter 3.3. The CTDs performed by the vessel survey, could be probably used to asses the level of no motion in the thermal wind approximation.*

R: Both CTDs performed by the vessel survey and those mounted on gliders can only provide relative geostrophic velocities via thermal wind equations. An estimate of the reference level to calculate the absolute geostrophic velocities would need: a) either a direct measurement of velocities like in Davis et al. (2008) and in Pickart et al. (2005), where depth-averaged velocities from glider drifts and vessel-mounted ACDPs were used, respectively; b) or an accurate sea surface slope field. Due to the limitations of satellite altimetry near the coasts, this latter approach is usually followed only in high-resolution modelling papers (e.g. Fraser and Inall (2018)). We explicitly cite the work by Davis et al. (2008) in the new version (page 7 line 7) and we stress out once again that sensitivity tests on $z_0$ between 500m and 700m show very limited variability (page 7 lines 9-10).

***Page 9 line 2 : The reference to the Beaufort scale seems to be not relevant in this context.***

R: we have rephrased the point (page 9 lines 9-11) clarifying that the chosen wind speed threshold (10 ms$^{-1}$) is consistent with previous Mistral observations in the Toulon area, such as Caccia et al. (2004) and Guénard et al. (2005).

***Page 10 line 15 : The reference to the Guibout Atlas is not sufficient. Please describe shortly the end of summer conditions and interpret the observed salinity minimum (present but not commented in Guibout).***

R: we have expanded the interpretation of the hydrographic transect including water masses dynamics specifying that the $\theta$ and $S$ values in Transect 1 show a strong thermocline at 50 m in the offshore central region, accompanied by a salinity minimum (S < 38 psu) right below the thermocline in the frontal zone. The fresher water centered around 80 m depth represents the core of the Modified Atlantic Water in the shallowest layer, laying over the Levantine Intermediate Water centered around a core with the highest salinity values (S > 38.55 psu) and a relative temperature maximum. The presence of stratification with a strong thermocline at 50-100 m, together with the salinity minimum right below it, are typical of late summer conditions (Guibout (1987) pp. 16 and 18 off Toulon and p. 36 off Nice, Albérola et al. (1995)) still present at the beginning of the experiment. Page 10 lines 20-26 of the new manuscript.

***Page 10 line 20 : date(1995) or indices(a) are missing in the references for this two citations. (Alberola, Petrenko)***

R: Albérola et al. (1995); Petrenko (2003) are listed in the references respectively on page 20 and page 23.

***Page 32 figure 8 : really cross-shore ? : along shore , cross section or zonal transport would be more appropriate.***

R: we have rephrased as suggested (page 33 of the new manuscript).

**References**

Albérola, C., Millot, C., and Font, J.: On the seasonal and mesoscale variabilities of the Northern Current during the PRIMO-0 experiment in the western Mediterranean Sea, Oceanologica Acta, 18, 163–192, 1995.

Berta, M., Griffa, A., Magaldi, M. G., Özgökmen, T. M., Poje, A. C., Haza, A. C., and Olascoaga, M. J.: Improved surface velocity and trajectory estimates in the Gulf of Mexico from blended satellite altimetry and drifter data, J. Atmos. Oceanic Technol., 32, 18801901, https://doi.org/10.1175/JTECH-D-14-00226.1, 2015.

Bouffard, J., Vignudelli, S., Cipollini, P., and Menard, Y.: Exploiting the potential of an improved multimission altimetric data set over the coastal ocean, Geophysical Research Letters, 35, https://doi.org/10.1029/2008GL033488, 2008.

Caccia, J.-L., Guénard, V., Benech, B., Campistron, B., and Drobinski, P.: Vertical velocity and turbulence aspects during Mistral events as observed by UHF wind profilers, Annales Geophysicae, 22, 3927–3936, https://doi.org/10.5194/angeo-22-3927-2004, 2004.

Davis, R. E., Ohman, M. D., Rudnick, D. L., and Sherman, J. T.: Glider surveillance of physics and biology in the southern California Current System, Limnol. Oceanogr., 53, 2151–2168, 2008.

Fraser, N. J. and Inall, M. E.: Influence of Barrier Wind Forcing on Heat Delivery Toward the Greenland Ice Sheet, Journal of Geophysical Research: Oceans, 123, 2513–2538, https://doi.org/10.1002/2017JC013464, 2018.

Gómez-Enri, J., Cipollini, P., Passaro, M., Vignudelli, S., Tejedor, B., and Coca, J.: Coastal Altimetry Products in the Strait of Gibraltar, IEEE Transactions on Geoscience and Remote Sensing, 54, 5455–5466, https://doi.org/10.1109/TGRS.2016.2565472, 2016.

Guénard, V., Drobinski, P., Caccia, J.-L., Campistron, B., and Bench, B.: An Observational Study of the Mesoscale Mistral Dynamics, Boundary-Layer Meteorology, 115, 263–288, https://doi.org/10.1007/s10546-004-3406-z, 2005.

Guibout, P.: Atlas Hydrologique de la Méditerranée, Laboratoire d'Océanographie Physique du Museum National d'Histoire Naturelle, 1987.

Kundu, P. K.: Ekman Veering Observed near the Ocean Bottom, Journal of Physical Oceanography, 6, 238–242, https://doi.org/10.1175/1520-0485(1976)006⟨0238:EVONTO⟩2.0.CO;2, 1976.

Petrenko, A. A.: Variability of circulation features in the Gulf of Lion NW Mediterranean Sea. Importance of inertial currents, Oceanologica Acta, 26, 323–338, 2003.

Pickart, R. S., Torres, D. J., and Fratantoni, P. S.: The East Greenland Spill Jet, Journal of Physical Oceanography, 35, 1037–1053, https://doi.org/10.1175/JPO2734.1, 2005.

Sammari, C., Millot, C., and Prieur, L.: Aspects of the seasonal and mesoscale variabilities of the Northern Current in the western Mediterranean Sea inferred from the PROLIG-2 and PROS-6 experiments, Deep-Sea Research, 42, 893–917, 1995.

Vignudelli, S., Cipollini, P., Astraldi, M., Gasparini, G. P., and Manzella, G.: Integrated use of altimeter and in situ data for understanding the water exchanges between the Tyrrhenian and Ligurian Seas, Journal of Geophysical Research: Oceans, 105, 19 649–19 663, https://doi.org/10.1029/2000JC900083, 2000.

[Figure]

Figure 1: Average phase angle (top panel) and magnitude correlation (bottom panel) between HFR total surface currents and ALADIN winds, estimated with the method by Kundu (1976). Negative angles indicate currents to the right of the wind direction. The limits of the selected wind events (E1, E2, and E3) are indicated with red dashed lines (see Fig.2b in the manuscript for more details).